# The Transformative Role of 3D Culture Models in Triple-Negative Breast Cancer Research

**DOI:** 10.3390/cancers16101859

**Published:** 2024-05-13

**Authors:** Xavier S. Bittman-Soto, Evelyn S. Thomas, Madeline E. Ganshert, Laura L. Mendez-Santacruz, J. Chuck Harrell

**Affiliations:** 1Department of Pathology, Virginia Commonwealth University, Richmond, VA 23284, USA; bittmansotx@vcu.eduthomases4@vcu.edu (E.S.T.); 2Massey Comprehensive Cancer Center, Virginia Commonwealth University, Richmond, VA 23284, USA; 3Division of Cancer Biology, University of Puerto Rico Comprehensive Cancer Center, San Juan, PR 00921, USA; 4Department of Biology, Loyola University Chicago, Chicago, IL 60660, USA; mganshert@luc.edu; 5Department of Biology, University of Puerto Rico-Rio Piedra Campus, San Juan, PR 00925, USA; laura.mendez1@upr.edu

**Keywords:** breast cancer, triple-negative breast cancer, 3D culture models, 3D culturing techniques, cell aggregates, spheroids, patient-derived organoids, organoids, extracellular matrix

## Abstract

**Simple Summary:**

This review article delves into the expanding role of 3D culture models in advancing breast cancer biology comprehension and enhancing drug response predictions, with an emphasis on triple-negative breast cancer (TNBC). It highlights the indispensable contribution of advanced technologies like patient-derived organoids and multi-omics analyses in unraveling tumor biology intricacies and drug sensitivity patterns, thereby fostering the creation of personalized therapeutic approaches. Additionally, it underscores the necessity of understanding the clinical translation for optimal model selection in TNBC therapy development, while acknowledging the impact of differences in cell culture models and culture conditions on drug response disparities.

**Abstract:**

Advancements in cell culturing techniques have allowed the development of three-dimensional (3D) cell culture models sourced directly from patients’ tissues and tumors, faithfully replicating the native tissue environment. These models provide a more clinically relevant platform for studying disease progression and treatment responses compared to traditional two-dimensional (2D) models. Patient-derived organoids (PDOs) and patient-derived xenograft organoids (PDXOs) emerge as innovative 3D cancer models capable of accurately mimicking the tumor’s unique features, enhancing our understanding of tumor complexities, and predicting clinical outcomes. Triple-negative breast cancer (TNBC) poses significant clinical challenges due to its aggressive nature, propensity for early metastasis, and limited treatment options. TNBC PDOs and PDXOs have significantly contributed to the comprehension of TNBC, providing novel insights into its underlying mechanism and identifying potential therapeutic targets. This review explores the transformative role of various 3D cancer models in elucidating TNBC pathogenesis and guiding novel therapeutic strategies. It also provides an overview of diverse 3D cell culture models, derived from cell lines and tumors, highlighting their advantages and culturing challenges. Finally, it delves into live-cell imaging techniques, endpoint assays, and alternative cell culture media and methodologies, such as scaffold-free and scaffold-based systems, essential for advancing 3D cancer model research and development.

## 1. Introduction

From basic cell aggregates to structures resembling organs, three-dimensional (3D) cell culture models have emerged as crucial tools in biomedical research for studying human diseases [1,2]. These systems accurately replicate numerous aspects of organs and patient tumors, including architecture, growth patterns, diversity, invasion tendencies, microenvironmental interactions, and response to drugs, offering valuable insights into disease complexities [3,4]. Scientists are currently developing 3D cell culture models from patient biopsies to improve existing research strategies and define novel therapeutic targets [1,5,6]. Among these models, 3D cancer models like patient-derived xenografts (PDXs) and PDX-organoids (PDXOs) show promise in understanding tumor biology and assessing treatment effectiveness in environments that are more similar to cellular structures that arise in patients as compared to traditional two-dimensional (2D) cultures [7].

Triple-negative breast cancer (TNBC) is aggressive and presents unique clinical challenges and limited treatment options due to the absence of the targetable estrogen receptor (ER), progesterone receptor (PR), and the human epidermal growth factor receptor 2 (HER2) [8]. Notably, chemotherapy remains the primary treatment strategy for TNBC patients, with only a few targeted therapies available [9]. Three-dimensional cancer models derived from TNBC tumors are proving to be reliable predictors, enhancing the translation of preclinical data [3]. However, these models pose challenges in their establishment, maintenance, scalability, and long-term viability [10].

This review underscores the significance of PDO and PDXO models in advancing our understanding of TNBC biology and precision medicine. It explores diverse 3D culture models and techniques, highlighting their advantages and applications. Additionally, it discusses promising technologies like microfluidic platforms and tissue engineering approaches, which could improve the fidelity and usefulness of 3D cancer models in TNBC research. In essence, the objective is to guide researchers and promote progress in TNBC research by offering insights into the optimal selection of culture models and experimental design.

## 2. Materials and Methods

This comprehensive examination relies on literature from 2016 to 2024, supplemented with pertinent studies from 2013 and 2015. We conducted searches on PubMed using keywords related to “Triple-negative breast cancer”, “3D cell culture models AND techniques”, “3D breast cancer models”, and “drug discovery” in diverse combinations. Approximately 34 studies emphasizing the application of 3D cancer models in triple-negative breast cancer research were selected through this process.

## 3. Results

### 3.1. Three-Dimensional Cell Culture Models

#### 3.1.1. Cell Aggregates and Spheroids

Cancer cells can assemble into 3D structures called cell aggregates or spheroids (Figure 1A) [11]. These models are considered simple 3D structures that resemble certain aspects of tumor architecture, including cellular communication and nutrient distribution [12]. Notably, previous studies have shown that migrating or circulating tumor cells (CTCs) have higher abilities to successfully form metastatic tumors when they cluster together [13]. By recapitulating some of the tumor’s features, including cell–cell interactions, growth kinetics, signaling mechanisms, metastasis processes, and drug response, cell aggregates are invaluable models for exploring diverse aspects of cancer biology [5]. However, they are often composed of a single-cell type, potentially limiting their ability to accurately reflect the genetic and molecular diversity of the patient’s tumor, including the intricate components of the surrounding microenvironment.

Moreover, 3D model formation efficiency varies among cancer cell lines, with some developing compact spheroids, while others form loose or tight cell aggregates. This phenomenon is largely influenced by differences in cell–cell adhesion properties, as evidenced by recent findings [12,14]. Cancer cells that form compact spheroids were shown to express elevated levels of E-cadherin, while those forming tight aggregates exhibit high expression of N-cadherin. This is particularly crucial, as studies have linked drug sensitivity to the size and shape of spheroids and cell aggregates [12]. For instance, Huand et al. reported that TNBC MDA-MD-231 spheroids exhibited increased resistance to antitumor compounds compared with the 2D culture [15]. Nonetheless, cell aggregate and spheroid models have been proven to be particularly useful in identifying potential anti-cancer treatments through drug screens in vitro, including for TNBC [16,17,18].

#### 3.1.2. Organoids and Tumoroids

Organoids are self-organizing multicellular 3D structures, derived from either stem cells or tissues, that faithfully mirror the architecture and functionality of organs or tissues [5,12,19]. Morphological observations have shown that most of the breast cancer organoids show cystic or solid phenotypes, with few cases displaying grape-like and flower-like morphologies [20]. For example, organoids derived from ductal carcinoma biopsy typically yield solid, coherent organoids, whereas lobular carcinoma tends to generate less cohesive structures. Bhatia et al. reported that their TNBC organoids exhibit a compact cell-sphere morphology compared to the acinar structure found in normal organoids [3]. By recapitulating the intrinsic characteristics of the tissue of origin, organoid models offer an innovative approach to investigating organ-specific diseases and developing therapeutic strategies.

Additionally, tumoroids or patient-derived organoids (PDOs) (Figure 1B), generated from primary tumor cells excised from patients, allow for the study of tumor development, growth kinetics, invasion dynamics, and drug response in a 3D complex [6]. Tumor microenvironment interactions can also be assayed by co-culturing organoids with other cell types such as stromal cells, vascular endothelial cells, or immune cells [21]. However, some limitations must be considered regarding organoid applications, including culturing challenges and the efficiency of replicating the tumor’s heterogeneity. Varying propagation capabilities among TNBC organoids have been observed, with some successfully propagated for more than 10 passages, while others experienced dropout from the culture. In this regard, low starting material, limited proliferation, and normal and stromal outgrowth are among the primary reasons for the short-term culture growth of PDO models. Interestingly, long-term cultured TNBC PDOs (up to 18 passages) have been shown to present remarkable morphological similarity to that of the original patient tumors when transplanted into NOD/SCID mice [3]. Overall, PDOs have proven to pose potential implications for personalized medicine approaches, while offering notable advantages in terms of both time and cost compared to animal models.

#### 3.1.3. Patient-Derived Xenografts (PDXs) and PDX-Organoids (PDXOs)

PDXs are one of the most commonly used cancer models for pre-clinical research (Figure 1C). In a PDX, cancer cells or tissue fragments directly obtained from a patient are implanted into immunodeficient mice, highly preserving the cellular and histopathological structure of the original tumor, including components of the tumor microenvironment [7,21]. These unique features enable large-scale genotype–response correlations, relevant therapeutic interventions, and subsequent tumor transplantation [6,19]. Additionally, PDX tumor cells that conserve the biological and molecular characteristics of the original tumor can be assayed for in vitro applications. Nonetheless, some limitations regarding PDX models include higher experimental costs and low engraftment success rates, with some requiring up to eight months to fully develop [22]. Notably, Guillen et al. reported a 29% successful rate in establishing TNBC PDX from biopsies that propagated for at least two generations [6]. Furthermore, after initial tumor growth in mice, some cell types do not expand or die, and usually, the selection of the more proliferative cell types arises during subsequent passing from mouse to mouse.

Organoids can be derived from PDX tumors (referred to as PDXOs, Figure 1D) and enable the study of tumor growth and motility ex vivo in both long- and short-term cultures [22,23]. PDXOs present a cost-effective and practical alternative to animal models and have revolutionized certain techniques associated with PDX studies, such as the detection of minute metastatic lesions through fluorescent imaging of PDXO cells [19]. Moreover, PDXOs have been instrumental in high-throughput drug screening, with reported results translational to PDXs, yielding promising outcomes in cancer research. Nonetheless, certain limitations of PDXOs as compared to PDXs must be considered, including differences in cellular composition (lack of stroma, blood vessels, neutrophils, etc.) and drug dosing differences that are inherent to controlled in vitro systems as compared to metabolic processes within an in vivo setting [4,6,7,21].

In summary, each 3D model has its advantages and limitations (Table 1), and the selection of the appropriate model depends on the nature of the cells and the specific research questions of the study.

### 3.2. Three-Dimensional Cell Culturing Techniques

#### 3.2.1. Scaffold-Free System

The scaffold-free method relies on cells’ ability to self-organize into 3D tissue-like structures without external scaffold material that cells can bind to (Figure 2A). Cells assemble into multicellular structures that resemble biological tissues through direct cell-to-cell interactions as well as environmental inputs. These models provide a flexible platform for researching tissue biology, disease processes, and responses to compounds. However, their variety and complexity can present difficulties regarding standardization and scalability in experimental settings.

##### Hanging Drop System

The hanging drop method utilizes gravity and surface tension to create 3D cell models that can be manipulated to study different modalities of cancer, including TNBC (Figure 2A1). Cell suspension drops are placed under the lids of non-coated culture plates, after which the entire plate is inverted, creating droplets. The cells within the culture medium aggregate at the tips of the droplets through gravitational force and are in direct contact with one another. With continuous cell proliferation and the deposition of the extracellular matrix, the compact and homogenous 3D cell model is formed at the medium/air interface. This method provides a reliable technique to create spheroids and organoids as it is easy to handle, inexpensive, and usually does not require long-term culture or special cell culture media. Increasing the viscosity of the suspension, by adding material such as methylcellulose (Methocel), allows for more regularly shaped spheroids. However, spheroids formed by this method are susceptible to disaggregation due to plate disturbance and evaporation. This method also requires a high level of technical expertise, as the changing of media and treatment of the spheroids can be quite difficult to do without disturbing the spheroid [31,32,33].

##### Magnetic Levitation System

Magnetic levitation utilizes a magnetic field to suspend cells and form spheroids and organoids (Figure 2A2). The cells of interest are first incubated overnight with a magnetic nanoparticle, such as Fe_3_O_4_, to allow the cells to attach to the nanoparticles. The cells are then detached and resuspended in a cell culture medium and placed in a well plate of choice [34]. A magnetic drive is then placed above the plate, emitting a magnetic field that causes the cells to levitate toward the air–liquid interface and aggregate to form 3D structures [35]. The cells then form their extracellular matrix, and cellular proliferation is seen in the magnetic levitation environment, even without magnetic nanoparticles, suggesting the cell’s innate proliferative ability [34]. The shape of the 3D culture is primarily determined by the magnetic attraction force, while concentration variations of magnetic nanoparticles have been observed to influence both the size and quantity of cells within the resulting 3D structure, with higher nanoparticle concentrations yielding larger structures.

##### Ultra-Low-Adhesion Plate Culture

Ultra-low-adhesion (ULA) plates utilize a U-bottom well which has a non-adhesive surface to force the cells to remain suspended and minimize cell–substrate adhesion (Figure 2A3). ULA plates are known for generating reproducible spheroids that streamline repeatability and are useful for high-throughput drug screening [36]. Poly-2-hydroxyethyl methacrylate (PolyHEMA)-coated plates are a commonly employed non-adhesive surface when utilizing ULA plates to allow for cell suspension maintenance and enhanced viability and functionality. PolyHEMA plates have been widely used in TNBC research and the efficacy of various cancer drug treatments [37,38].

##### Agitation-Based Culture

Spheroids can also be produced by continuously agitating a cell suspension, which allows for more cell-to-cell interaction and prevents cells from adhering to the walls (Figure 2A4). This method utilizes different types of bioreactors that are responsible for continuously stirring the suspension. Two of the most common bioreactors used include the spinner flask and the rotary cell culture system. The spinner flask consists of a three-neck flask and a magnetic stirring element that can continuously stir the suspension while allowing for gas exchange and waste removal. This method, while readily accessible and easy to produce, can lead to physiological changes in the cell due to the high shear stress related to the liquid movement due to the force of the stir bar. The rotary cell culture system allows for less shear stress on the cell suspension by rotating the whole vessel rather than implementing a force-inducing stir bar. The vessel mimics a microgravity environment and is rotated slowly to allow for cell aggregates to form, after which the rotational speed increases to maintain spheroid formation [32,39,40].

#### 3.2.2. Scaffold-Based Systems

In scaffold-based models, cells are arranged into three-dimensional structures that mimic the architecture of actual tissues by using external materials to mimic the extracellular matrix (Figure 2B). These scaffolds enable the development of intricate tissue-like structures with distinct architectures by providing tight control over cell organization, matrix composition, and mechanical attributes. Furthermore, scaffold-based models provide a controlled setting for investigating drug reactions, signaling pathways, and interactions between cells and matrices.

##### Polymer Scaffold-Based Culture

Polymer scaffolds provide a biomimetic environment that mimics the natural extracellular matrix and effectively supports cell attachment and proliferation with great precision, allowing for better cell signaling throughout tissue development (Figure 2B1). Polymer scaffolds can either be naturally, occurring from natural polymers, or synthetically derived, the latter of which can be made to have specific mechanical and biochemical properties [41,42]. A study conducted by Rabie et al. utilized electrospun chitosan/polyethylene oxide nanofibrous scaffolds with different concentrations of chitosan (C2P1 and C4P1) to test its effect on the formation of 3D breast tissue scaffolds and found that it was not only reliable and cost-effective, but also spontaneously fused to form 3D microtissues without implementing any external growth factors after 10 days of culture in the scaffold containing a lower chitosan content (C2P1) [42].

Previous studies have shown the benefits of nanofibrous scaffolds in promoting the formation of mammospheres but have also shown them to be unreliable for high-throughput methods and to promote spheroid instability; however, Rabie et al. found that C2P1 not only produced a high number of spheroids but also demonstrated high viability. Similar results have been observed with synthetic polymer scaffolds, which have been shown to have improved cell-to-cell and cell-to-matrix interactions as well as increase successful replications of the molecular characteristics associated with epithelial–mesenchymal transition properties [41,43].

##### Hydrogels Scaffold-Based Culture

Hydrogel scaffolds are a subset of polymer scaffolds that have a more specialized usage, as they are recognized for their ability to absorb and retain large amounts of water and gel-like consistency (Figure 2B2). The most widely used hydrogel scaffolds include Matrigel and Cultrex, considered to be basement membrane extracts (BMEs). BMEs are a mixture of laminin, type IV collagen, entactin, proteoglycans, and growth factors, secreted by Engelbreth–Holm–Swarm mouse sarcoma cells [44]. The proportion of hydrogel to media within these cultures can have impacts on the studies that can be performed and the physiology of the cells [44]. A study performed by Badea et al. found that Matrigel allowed for TNBC MDA-MB-231 models that portrayed uniform morphology, increased diameter, good circularity, and increased expression of a proliferation marker compared to those grown without Matrigel. More specifically, there was higher compaction of cells in spheroids cultured in 2.5% Matrigel, whereas cells grown in the absence of it were not able to form spheroids, only tight cell aggregates. Moreover, the 5000 cells cultured in Matrigel reached a cellular size of 700 μm after the sixth day of culture, while the latter only reached 350 μm in size [45].

Despite their ability to promote viability and morphology, these common hydrogel scaffolds can pose certain consequences, including batch-to-batch variability, and limited ability for mechanical modifications, which are vital for developing an understanding of the function of mechanical and structural signals supplied by the tumor microenvironment in cancer growth and tumor response to drugs. Notably, Prince et al. implemented a novel approach using a biomimetic nanofibrillar hydrogel (EKGel) with controllable stiffness to grow TNBC PDOs and found it to have enhanced covalent crosslinking for stability and fibrous structure that mimics the architecture of the tumor extracellular matrix [44].

##### Decellularized Extracellular Matrix (ECM) Culture

Decellularized ECM (dECM) refers to cells being removed from tissues and organs, thereby isolating the ECM structure and preserving its architecture (Figure 2B3). ECM provides a physiologically relevant scaffold, as it maintains the fibrillary 3D network and contains the pure major proteins, including collagen, laminins, proteoglycans, and cell growth factors that more accurately mimic breast cancer cells in vitro. dECM also helps regulate tissue homeostasis and angiogenesis, morphology, structure, gene expression, and cell signaling with physiological significance. There are three distinct methods for decellularization: physical, chemical, and enzymatic. Each method can have varying effects on the ECM components, so it is important to differentiate which method is most optimal for the specific line of TNBC cells used in the experiment [46,47].

##### Tumor-on-a-Chip Culture

Tumor-on-a-chip is a specific application of microfluid technology designed specifically to study tumors and mimic all the cellular, biological, and structural features of a true TNBC cell (Figure 2B4). It forms compact spheroid complexes in the form of tissue structures that display relevant cell–cell and cell–matrix interactions. It replicates the fluid flow, continuous perfusion, shear stress, nutrient supply, and waste removal found in vivo and can more accurately represent tumor pathogenesis. The growth and progression of TNBC are highly linked to the tumor microenvironment in which it grows, as it assists with the initiation, development, invasion, and metastasis of these cells [48].

Lanz et al. focused on TNBC cells and how tumor-on-a-chip models responded to anti-cancer drug treatments in comparison to 2D models. They found that these models provided both a more physiologically accurate response and enabled the simultaneous culture of 96 perfused microtissues, beneficial for high-throughput screening where limited patient biopsy material and the need to screen numerous compounds and combinations for effective therapies pose challenges [49].

##### Long-Term Culture

Breast cancer organoids and PDXOs are the most suitable for long-term culturing, as they allow for a better representation of the complexity of breast cancer’s genotypic and phenotypic heterogeneity, which thereby facilitates better results for observing tumor development, heterogeneity, and metastasis (Figure 2B5). Breast cancer cell lines, including TNBC cells, have a slower proliferation rate compared to other cell lines, with the splitting ratio varying from 1:6 weekly to 1:2 biweekly or even less frequently. Therefore, the culturing mechanism must be optimized to promote enhanced proliferation and survival. Notably, Dekkers et al. described an optimized experimental protocol for long-term culturing and xenotransplantation of breast cancer organoids [50]. Their study compared two types of expansion media, Type 1 being an already established protocol for breast cancer organoids, and Type 2, which was developed for the long-term culturing of ovarian cancer organoids. Despite minor differences in additives, the base media remain similar, with the main components being R-spondin 1, noggin, B27 + VitA, nicotinamide, N-acetylcysteine, primocin, Y-27632, heregulin B1, FGF-10, A83-01, and EGF. Various studies confirm the benefits of the above additives, including Urbischek et al., who confirmed that minimizing serum impurities was important for synthetic matrices [51]. They found that bacterially derived preparations of R-spondin 1 and Gremlin-1 and BMP antagonists like noggin are essential growth factors for culturing organoids derived from epithelial tissue and are free of impurities such as serum growth factors (FBS) that may hinder organoid growth and reproducibility.

To maintain organoids for long-term culture, Dekkers et al. recommend passaging 7–21 days after organoid establishment or after the previous split, at a 1:2–1:8 ratio, with the ratio being determined by confluency and the passaging being just before the cells centralized at the BME drop start dying [50]. Liu et al. conducted a study comparing supplemental organoid maintenance protocols for MDA-MB-231, MCF-7, and BT-474 in both static and perfused 3D cell culture platform, using the PerfusionPal system and SeedEZ scaffold—a glass microfiber-based scaffold designed for long-term cell growth and repeat-dose drug testing [52]. The study found that perfusion enhanced the culture environment’s homeostasis and raised cell metabolic activity and proliferation for longer periods, with cell growth considerably higher compared to static conditions by day 7 and continuing to develop through day 21. Although PerfusionPal and SeedEZ are ways to implement perfusion without products derived from animals, perfusion can also be observed with xenotransplantation of tumors into mice and further emphasize the PDXO’s functionality for long-term culturing.

#### 3.2.3. 3D Culturing Challenges and Alternative Approaches

Proteinaceous matrices like collagen or Matrigel are often used in 3D cell culture but may produce unintended signaling due to excess growth factors, limiting their mimicry of primary tumors. Nayak et al. developed an alternative scaffold of biodegradable poly (ε-caprolactone) (PCL) for studying breast metastasis. However, isolated primary breast cancer cells failed to attach and proliferate on this scaffold due to the absence of cell-adhesive ligands in PCL. Instead, they pre-cultured breast cancer-associated fibroblasts on the scaffold, allowing them to deposit ECM where cells can adhere, allowing tumor formation. This approach holds promise for more accurate modeling of breast cancer complexities in vitro [25].

Moreover, conventional 3D cell culture models typically generate tumors from a large pool of cells, presenting challenges in identifying specific phenotypes driving aggressive behavior at a single-cell level. Jain et al. pioneered a deterministic 3D culturing approach that enables the cultivation of individually selected cells in relative isolation. By employing a glass capillary-based microfluidic system, they extract single cells from a population and seed them on top of individual collagen domes. This approach not only facilitates the identification of samples with actively proliferating cells but also enables the isolation and further characterization of tumor-promoting cells within a sample. Notably, they tested this approach by seeding single cells from two breast cancer models, including MCF-7 (poorly aggressive, luminal), MDA-MB-231 (highly aggressive, triple negative), and Caco-2 colorectal cancer (poorly aggressive). Remarkably, both poorly aggressive cell lines develop tumors with morphologies reminiscent of their cancer subtypes. However, none of the single cells from TNBC formed tumor masses; instead, they displayed their aggressive nature through migratory behavior. In essence, this innovative method facilitates the study of the progression of single-cell-derived tumors and enables the transfer of tumors for long-term culture and downstream analyses [26].

### 3.3. Three-Dimensional Culturing Media and Additives

#### 3.3.1. Base Media for 3D Cell Culture

Recent studies have successfully established PDO and PDXO models from diverse breast cancer subtypes, including TNBC, effectively preserving essential tumor features [3,6]. However, the process of developing and maintaining these models poses significant challenges, sometimes requiring specialized expertise. The choice of culturing media plays a crucial role in maintaining cell preservation and proliferation within 3D models. While traditional additives (Table 2) are being employed for TNBC organoid culture, ongoing advancements aim to enhance the formation of 3D models and address associated limitations. Notably, Mazzucchelli et al. introduced a pioneering protocol for generating patient-derived organoids from breast cancer samples, including TNBC cases [53]. Their protocol emphasizes Neuregulin 1 and the Rho-protein-dependent kinase (ROCK) inhibitor Y-27632 as crucial supplements for organoid establishment and long-term expansion, while caution is advised with elevated EGF concentrations due to potential structural disruption. However, their study only included one TNBC organoid model among the 21 breast cancer organoids generated.

Similarly, Sachs et al. developed a vigorous protocol enabling the culturing of human mammary epithelial organoids, facilitating the generation of primary and metastatic breast cancer organoid lines [4]. Notably, their method excludes mesenchymal cells and relies on key factors such as EGF, R-spondin, Noggin, and BME, with additional additives tailored to specific requirements. Other expansion media have been reported for TNBC organoid culture, which includes supplements such as N-Acetyl-L-Cysteine, Nicotinamide, B-27, FgF-10, SB202190, and RSP01, and have underscored the significant challenges associated with the aggressive and genetically unstable nature of TNBC [4,30,57,58].

##### Plasmax Culture Media

Recent studies have shown that Plasmax culture media allows cancer cells to mimic tumorgenicity more closely and the metabolic function of TNBC cells. Plasmax is a physiologically relevant culture media that closely resembles the nutrient composition of human plasma and contains 66 organic components as well as growth-enhancing elements such as vanadium, zinc, manganese, copper, and selenium [59]. A study by Voorde et al. compared TNBC cell lines cultured in Plasmax versus the typical DMEM-F12 to observe which more closely resembled orthotopic xenografts in mice [59]. The study showed that Plasmax not only maintained cell growth, but also increased cell colony formation. More specifically, the sodium selenite in Plasmax, which provides selenite supplementation in addition to the selenite present in FBS, promotes colony formation, allows for high GPX4 activity in TNBC cells, and helps maintain survival and proliferation. TNBC spheroids grown within Plasmax also behaved more similarly to the orthotopic xenografts in comparison to cells grown within DMEM, with spheroids growing in the Plasmax resembling the metabolic profile of the xenografts more closely than within DMEM with just four days of culture. Furthermore, Plasmax allows for better reproducibility in comparison to historic cell culture serums, such as FBS, as the concentration of essential elements can be standardized and are not variable between batches, and trace elements, such as zinc, iron, and selenium can be supplemented in appropriate concentrations [60].

#### 3.3.2. Rho-Protein-Dependent Kinase (ROCK) Inhibitor

Rho-protein-dependent kinase (ROCK) inhibitor Y-27632, an ATP competitive inhibitor of ROCK, has been increasingly used as an additive for culturing organoids and PDXOs due to their beneficial effects in supporting cell viability, promoting cell proliferation, inhibiting apoptosis, and preserving the genotypic and phenotypic cancer cell traits, allowing 3D cultures to have more physiologically relevant characteristics [61]. Bhandary et al. found that the use of ROCK inhibitors allowed for increased microtentacle formation in detached breast cancer cells, which can help promote cell survival and may also provide better cell-to-cell interactions and aggregation [62]. However, Guillen et al. found that despite its necessity for optimal culturing, the use of Y-27632 may interfere with in vitro drug treatments, leading to skewed results when observing drug therapeutics [30].

### 3.4. Approaches to Assessing Growth Kinetics, Migration, Morphology, Autophagy, and Cell Death for PDO/PDXO

#### 3.4.1. Live Cell Imaging Techniques and Analysis Instruments

Live cell imaging techniques enable real-time monitoring of organoid growth dynamics and morphology. Optical imaging has proven to be highly effective for investigating organoid morphology and behavior, capitalizing on its compatibility with the model’s compact size. Its tissue-penetrating ability enables comprehensive cell measurement, while its non-invasive nature allows for continuous evaluation of viability, morphology, and heterogeneity over time. Various optical imaging techniques, including bright-field microscopy, light microscopy, phase microscopy, fluorescence microscopy, confocal microscopy, time-lapse microscopy, multiphoton fluorescence imaging, fluorescence lifetime imaging, and optical coherence tomography, are employed for these investigations [63].

Moreover, Walsh et al. demonstrated the utility of optical metabolic imaging (OMI) in studying cellular metabolism and drug responses within organoids, offering precise insights for personalized cancer treatment. OMI employs fluorescence microscopy and specific probes to directly observe metabolic processes like glycolysis and oxidative phosphorylation, offering real-time insights into cancer cell behavior and therapy responses. Additionally, cancer cells can be engineered to express fluorescence markers, providing valuable insights into nutrient utilization, energy generation, and response to stimuli, including drug treatments [64].

In live cell imaging, researchers employ cutting-edge instruments like Agilent/BioTek’s Cytation Imaging Readers and Sartorius’ Incucyte Live Cell Analysis System to study dynamic cellular processes in real time. These systems integrate digital microscopy with microplate detection, enabling high-resolution imaging and analysis directly within microplates. Moreover, they provide continuous live cell imaging within standard incubators, facilitating seamless tracking and analysis of dynamic cellular behaviors. Additionally, while many articles lack full details on how their images are obtained, other instruments worth mentioning are the Nikon Ti Eclipse Series for live cell fluorescence microscopy and Zeiss LSM Series Confocal Microscope for high-resolution visualization [65,66,67].

#### 3.4.2. Endpoint Assays

A variety of assays are utilized to evaluate migration, growth kinetics, and cell death within PDOs and PDXOs, providing valuable insights into their behavior and response to stimuli. Migration assays, such as transwell migration assays, measure the ability of cells to migrate through a porous membrane, offering insights into the invasive potential of organoids [68]. Growth kinetics can be assessed using proliferation assays, including cell counting, bromodeoxyuridine (BrdU) incorporation assays [69], and CellTiter-Glo assays [70,71], quantifying cell proliferation and viability over time. Notably, a recent study reported that CellTiter-Glo analysis on metastatic colorectal cancer (mCRC) PDOs was comparable with CyQUANT Cell Proliferation assay measurements and did not affect the correlation with patient response [72].

To evaluate cell death, assays such as Annexin V staining, propidium iodide staining, and TUNEL assays can be utilized to detect apoptosis and necrosis within organoids [73]. Furthermore, functional assays like caspase activity assays provide mechanistic insights into cell death pathways. Combined, these assays offer a comprehensive understanding of organoid behavior under various experimental conditions, aiding in the elucidation of underlying mechanisms and the development of targeted therapeutic strategies.

Additionally, assaying endpoints in 3D cancer models can be performed through histological and immunohistochemical (IHC) techniques. This process involves 3D culturing under optimal conditions, following treatment, fixing (e.g., formalin), and embedding for histological processing. Paraffin wax is the most widely used embedding medium due to its compatibility with routine histological techniques and ease of sectioning. Thin paraffin sections are stained with Hematoxylin and Eosin (H&E) to visualize tissue morphology and architecture, while IHC measures protein expression (e.g., Ki67). Notably, IHC involves antigen retrieval, primary antibody incubation, and visualization of antigen–antibody complexes. Through high-resolution imaging software, parameters such as tumor size, cell proliferation, and protein expression levels can be measured and quantified [15,74,75]. Interestingly, researchers are incorporating artificial intelligence (e.g., MOrgAna) and computational methods (e.g., Cellos) in organoid analysis to overcome the challenges of histological and IHC techniques, including overlap, over-illumination, or partial obscurement by noise [76,77].

### 3.5. PDO and PDXO Multi-Omics Analysis

#### 3.5.1. Two-Dimensional vs. Three-Dimensional Culture Multi-Omics Analysis

Omics studies on breast cancer organoids involve the comprehensive analysis of various biological molecules and components, such as genomics, transcriptomics, proteomics, and metabolomics, to gain insights into the molecular characteristics and behaviors of these models. Notably, organotypic cultures in breast cancer cells have been shown to significantly impact the miRNA transcriptional program, leading to changes in gene and protein expression, as well as cell morphology. A recent study on TNBC cell line Hs578T revealed 354 differentially expressed miRNAs in 3D culture relative to 2D. These transcriptional changes were associated with the regulation of key biological processes and signaling pathways, such as catabolic processes, hypoxia response, and focal adhesion. Interestingly, miR-935 was found to be downregulated in Hs578T 3D organotypic cultures compared to 2D and in TNBC patients compared to adjacent normal tissues, indicating its potential as a target in TNBC progression. The study also revealed distinct miRNA profiles in basal Hs578T cells compared to luminal MCF-7 and TNBC MDA-MB-231 cells, highlighting the specificity of miRNA expression to each culture in 3D [78,79,80].

#### 3.5.2. Omics Analysis Guides Model Selection

RNA-seq data from a recent research study identified MDA-MB-415, BT483, and EFM192A as the top three representatives of Luminal A, Luminal B, and HER2-Enriched subtypes, respectively. By leveraging transcriptomic and multi-omics analyses, they evaluated and compared organoid models to cell lines and human tumors, thereby facilitating the selection of the most suitable model for a specific tumor subtype. Interestingly, they identified the organoid model MMC01031 to show the highest transcriptome similarity and higher correlation with MET500 breast cancer samples than the corresponding most significant cell lines. Moreover, the organoid model W1009 demonstrated the highest transcriptome similarity and expression correlation with Basal-like MET500 breast cancer samples, surpassing even the HCC70 cell lines, renowned as the most significant TNBC cell line for the basal-like subtype [81].

Moreover, a recent study introduced a novel bioinformatic tool named Congruence Analysis and Selection of Cancer Models (CASCAM), which identified HCC1599 cell line-derived organoids as the most congruent model for TNBC, while BT549 organoids were found to be the least congruent model, particularly concerning WNT signaling. Notably, their study’s database only contained four PDO samples that clustered as TNBC. Overall, these findings collectively advocate for the use of organoids as a faithful reflection of patient biology, particularly advantageous for exploring cancer processes and gene expression-driven drug discovery [82].

### 3.6. Drug Discovery Using 3D Cancer Models

Therapeutic screens and drug testing using organoid models have revolutionized general understanding and approaches to cancer treatment. Organoid-based systems offer unparalleled precision in assessing drug responses and surpass traditional cell-based models [57,83]. Recent studies focusing on breast cancer have exemplified the efficacy of PDOs in evaluating drug responses, particularly in TNBC, by using these 3D models to assess changes in cell viability and synergism. Specifically, organoids provide a more accurate prediction of drug responses observed in clinical cases by representing tumor heterogeneity and mutations throughout long-term cultures; for instance, Chen et al. demonstrated this accuracy upon finding a strong correlation between organoid pharmaco-phenotyping and clinical outcomes [20]. Furthermore, a recent study by Campener et al. revealed a response in TNBC organoid viability to docetaxel and tamoxifen [84]. PDOs have also been instrumental in evaluating novel compounds like iPR, an iRGD peptide conjugated with a bromodomain-containing protein 4 (BRD4) proteolysis-targeting chimera (PROTAC) through a GSH-responsive linker. In the study by He et al., PDOs serve as an important preclinical model for assessing the effects of anticancer drugs and depth of tissue penetration that cannot be evaluated using cell line procedures [85].

The use of PDOs in combination therapies has also proven effective. One such example includes Tan et al. elucidating the synergistic effect of IN10018, an inhibitor of focal adhesion kinase (FAK), and crizotinib, an inhibitor of ROS proto-oncogene 1 in TNBC organoids. Specifically, this combination reduced cell viability by 70% by inhibiting proliferation, enhancing apoptosis, and inducing ferroptosis through the associated increase in p53 levels [86]. The study of synergistic effects using PDOs is also seen in the work of Rowdo et al. in verifying the synergistic properties of temozolomide, a DNA damaging agent, and talazoparib, a Poly (ADP-ribose) inhibitor (PARPi) to target mutant p53 (mtp53) signaling. Moreover, mtp53 and PARP proteins may serve as important biomarkers and prognostic indicators for TNBC patients who may benefit from precision medicine treatments such as temozolomide and talazoparib combination treatments [87]. These findings underscore the utility of patient-derived TNBC organoids in advancing personalized therapy strategies.

While PDOs are widely used in cell viability experiments, this culture model also enables efficient and high-throughput screening that further supports the practice and capacity of precision treatments. This is essential for accurately predicting clinical responses, such as the work of Guillen et al. in identifying the highly effective FDA-approved drug erubilin through screening of TNBC PDX and PDXO models [6]. Additionally, high-throughput screening was essential to Rao et al. when screening 169 epigenetic compounds using TNBC PDOs, ultimately determining the anti-tumor activity of panobinostat (targeting histone deacetylase), pacritinib (targeting JAK/STAT), TAK-901 (targeting histone demethylases), and JIB-04 (targeting aurora kinase pathways) [88]. Önder et al. also utilized metastatic BC patient-derived organoids as a screening platform for antigen expression patterns to demonstrate the feasibility and flexibility of precision immunotherapy. Specifically, these studies demonstrated the use of T cell therapies that incorporated a chimeric antigen receptor (CAR) with adapter molecules (Ams) and adapter CARs (AdCARs) to generate Adapter CAR-T cells (AdCAR-T) and biotinylated monoclonal antibodies to target specific antigens on the tumor cell surface and facilitate lysis. Throughout this work, the use of PDOs was necessary to preserve receptor statuses and hotspot mutations throughout numerous passages [89].

In addition to the use of PDOs for testing and screening drug capabilities, recent studies showcase the use of PDOs following radiation or neoadjuvant chemotherapy (NAC), respectively. For instance, Shu et al. showed consistency between TNBC patient responses and organoid reactions to chemotherapy [90]. Additionally, findings by Pellizzari et al. indicated that the combination of Polo-like Kinase 4 (PLK4) and radiotherapy (RT) has a synergistic effect in decreasing organoid formation because of centriole overduplication and genomic instability [91]. Furthermore, Derouane et al. used TNBC PDOs to characterize tumor cell metabolism following NAC, concluding that glycolytic activity is increased in chemoresistant TNBC cells. To target this diminished sensitivity to NAC in cells surviving treatment, TNBC PDOs were then used to evaluate a combinatory treatment with paclitaxel and 2-deoxyglucose, a glucose analog to block hexokinase 2 activity and prevent metabolic adaptation. Such combination studies reduced the 2D and 3D growth capacity of TNBC, therefore indicating that glycolysis-interfering therapeutics may facilitate the improvement of TNBC patient responses to NAC [92].

Organoid models also shed light on small extracellular vesicles (sEVs) through studies conducted by Shen et al. aimed at characterizing the transcriptomic and proteomic profiles of sEVs from tumor organoids (O-sEVs) along with other sources, including paired normal tissue and plasma [93]. These findings provide insights into the complex tumor microenvironment and potential diagnostic and therapeutic implications. Specifically, 37 proteins were found to be upregulated in O-sEVs compared with sEVs derived from paired normal tissue, with no overlap with the 467 previously defined cancer-specific sEV proteins [93].

Overall, therapeutic screens and drug testing using organoid models offer a transformative approach to cancer research and treatment, providing enhanced precision, translational relevance, and a deeper understanding of tumor biology and therapeutic responses.

## 4. Discussion

In the realm of advanced breast cancer treatment, particularly in challenging cases like TNBC where effective therapies are scarce, 3D culture provides a real-time platform for exploring personalized treatment approaches and improving patient outcomes. Patient- or cell line-derived 3D cancer models bridge the gap between simplified 2D models and organismal models that are more expensive and inaccessible to many imaging or high-throughput methods. PDO-based drug profiling has not only identified effective agents for TNBC models in vitro but has also correlated with positive patient responses in clinical studies. Similarly, Guillen et al. identified birinapant as a potent antitumor agent for TNBC, while inhibiting lysyl oxidase overcomes chemotherapy resistance in PDXOs [6]. Hence, PDO and PDXO models can be positioned as complementary to PDX models in the drug discovery research field, offering the opportunity to leverage 2D model drug screens for initial drug selection, followed by rigorous testing in PDOs and PDXOs. Subsequently, PDOX derivation can be employed for in vivo validation studies, ensuring a comprehensive evaluation of drug efficacy.

Moreover, PDO and PDXO models can be derived from many breast cancer tissues, including specimens from heavily treated patients and rare subtypes. TNBC PDOs recapitulated the tumor-intrinsic properties of the original tumor at genomic, transcriptomic, and morphologic levels, making them an accurate in vitro model to study tumor biology. However, while PDOs typically retain the major cells of the mammary epithelium, Bhatia et al. reported that certain TNBC PDOs may exhibit a loss of lineage specificity, which may alter the heterogeneity and morphology of the models. These PDOs are often enriched for luminal progenitor (LP)-like cells, showing gene signatures reminiscent of aggressive MYC-driven, basal-like breast cancers. Additionally, they may display altered transcriptional patterns compared to the patient, such as the expression of luminal marker EPCAM^+^ and basal marker CD49f^+^. Hence, these TNBC organoids tend to be highly proliferative and undifferentiated, lacking significant cellular organization when compared to normal and luminal organoids, which typically exhibit less than 10% of proliferating cells [3].

Furthermore, alternative 3D culture techniques have revolutionized our understanding of aggressive cellular behavior in 3D breast cancer cultures, particularly in the context of TNBC. In contrast to traditional methods where tumors are generated from a large pool of cells, isolating single cells within a 3D environment enables precise identification of specific phenotypes driving aggression. The alternative approach proposed by Jain et al. for culturing isolated single cells in 3D (previously described in the 3D culturing section) allows for the characterization of the unique aggressiveness of TNBC at a single-cell level, offering valuable insights for targeted therapeutic strategies. Overall, the ability to derive models from patient-derived cells and develop 3D cancer models, such as PDO and PDXO, has significantly advanced TNBC research by providing a physiologically relevant platform for addressing drug discovery, cancer biology, and bioengineering, as extensively described in this article.

## 5. Conclusions

Transitioning from conventional 2D cell cultures to more complex 3D models poses significant challenges, especially within the field of breast cancer research. However, overcoming these hurdles and exploring innovative avenues is crucial for harnessing the full potential of 3D cancer models in precision medicine and drug discovery. This endeavor requires interdisciplinary collaboration among researchers, engineers, and clinicians to address technical complexities and advance breast cancer research effectively. Such collaborative efforts are pivotal for translating research findings into clinical practice and realizing the full potential of 3D models in improving outcomes for TNBC patients. Replicating the intricate tissue microenvironment in these models remains formidable, demanding advanced engineering techniques and precise control over cultural conditions, often at high cost. Bioprinting technologies offer promise in overcoming these challenges by enabling the creation of detailed 3D structures and enhancing reproductivity and consistency.

Nevertheless, translating differential gene and protein expression between 2D and 3D cultures into drug sensitivity poses uncertainties specific to TNBC research, and comparisons between these models may reveal discrepancies, potentially compromising study fidelity. Despite these challenges, promising directions for future research are emerging, such as the direct derivation of PDOs from circulating tumor cells (CTCs), offering invaluable insights into tumor heterogeneity, disease progression, and monitoring cancer burden, albeit with limitations due to low CTC yield.

Overall, each model—2D or 3D—has its advantages and limitations, and the selection of the appropriate model and culturing methodology depends on the nature of the cells and the specific research questions and goals of the study.

## Figures and Tables

**Figure 1 cancers-16-01859-f001:**
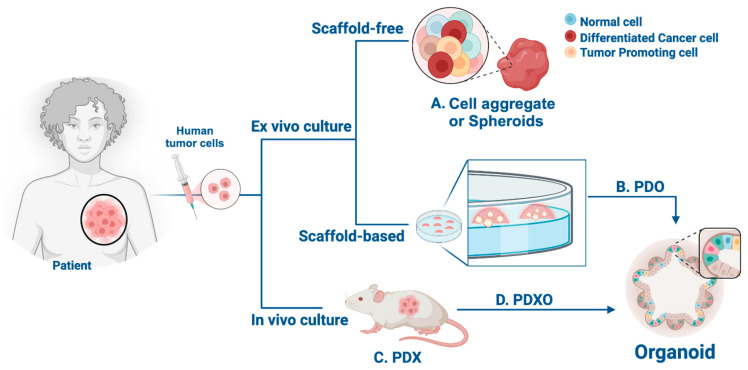
Schematic representation of epithelial tissue and novel 3D cell culture models for studying human tumor cells. Human tumor cells undergo ex vivo and in vivo culture techniques, including scaffold-free and scaffold-based systems. (**A**) Tumor cells from cell aggregates or spheroids represent a more physiologically relevant architecture compared to traditional 2D models. More complex 3D culture models include (**B**) patient-derived organoids (PDOs), (**C**) patient-derived xenografts (PDXs), and (**D**) organoids developed from patient-derived xenograft models (PDXOs). This figure is presented for the first time and was created with BioRender.com.

**Figure 2 cancers-16-01859-f002:**
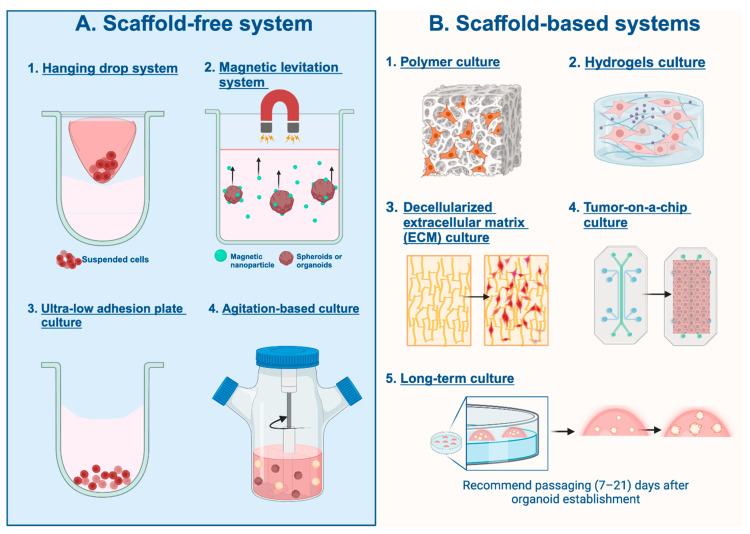
Schematic representation of various scaffold-free vs. scaffold-based 3D cell culture systems and their impact on cell behavior. (**A**) Scaffold-free systems include the hanging drop system (**A1**), magnetic levitation system (**A2**), ultra-low adhesion plate culture (**A3**), and agitation-based culture (**A4**). (**B**) Scaffold-based systems include polymer (**B1**), hydrogel (**B2**), decellularized extracellular matrix (**B3**), tumor-on-a-chip (**B4**), and long-term (**B5**) culture systems. This figure is presented for the first time and was created with BioRender.com.

**Table 1 cancers-16-01859-t001:** Advantages and challenges of various 3D cell culture models.

Cell Culture Models	Advantages	Challenges	References
2D cultured cell	Simple setup, well-established protocols, and low-cost maintenance. Easily scale up for high-throughput drug screens and offer a uniform environment for straightforward microscopy and immunostaining analysis.	Monolayer culture (1) alters cellular behavior, gene expression, and intricate cell interactions, (2) leads to discrepancies in drug responses, and (3) lacks modeling of spatial gradients, such as nutrients, oxygen, and signaling molecules.	[24,25]
3D cell aggregates and spheroid	Facilitate cell–cell interactions for studying complex cellular behaviors and signaling pathways. Enables a more accurate representation of cancer progression and drug response compared with 2D. Incorporating extracellular matrix components creates a biomimetic microenvironment conducive to cell adhesion, migration, and differentiation.	Often composed of single cell type, limiting to the reflection of molecular diversity. Variability in cell density, size, and shape complicates standardization, high-throughput screens, and imaging analysis. Non-adherent culture techniques may fail to represent tumor formation in vivo. Requires moderate-cost maintenance, specialized protocols, and complex equipment.	[26]
Organoid and PDO	Stem cell- or patient-derived organoids preserve histological, transcriptional, and genetic characteristics of the original tumor, including tumor heterogeneity and mutation patterns during long-term culture. Enables personalized medicine and facilitates drug response studies. Cost-effective compared with animal models and compatibility with emerging technologies.	Lack of standardized protocols for generation and expansion, leading to experiment variabilities. Cellular heterogeneity and uniformity affect maturation and stability, impacting reproducibility and data interpretation. Vascularization absence limits size and organ functions accurately. Requires costly and time-intensive maintenance.	[20,27]
PDX	Accurately represents in vivo biology by preserving tumor microenvironments and patient tumor characteristics. Excels in predicting drug responses, aiding therapy development, and enabling long-term studies of tumor behavior. Allows tumor growth and passage to create cohorts or cryopreserving to establish living tissue biobanks.	Technically demanding, requiring specialized equipment, facilities, and expertise in xenotransplantation. Low engraftment success rates, long generation cycles, and costly and time-consuming maintenance. Limited study of tumor–immune interactions due to the use of immunodeficient hosts.	[28,29]
PDXO	Faithfully mimics patient tumor traits, enabling predictive drug screening, long-term monitoring, and personalized medicine approaches. Co-culture options facilitate tumor–immune interaction studies.	Technically demanding and costly, requiring specialized equipment, facilities, and expertise in xenotransplantation and organoid culture techniques. Variable culture success rates and potential biases towards aggressive cell lines. Genetic and phenotypic changes over time due to adaptation of the host’s environment.	[30]

**Table 2 cancers-16-01859-t002:** Base culturing media and additive for 3D cell culture.

Additive	Description	References
Fetal Bovine Serum (FBS)	A universal natural growth supplement of tissue and cell culture media that contains growth-enhancing factors (growth factors, hormones, nutrients, etc.). Note: Smaller molecules within FBS are not fully understood, and their effects on cell cultures are not known and could lead to discrepancies in the reproducibility of cell cultures. FBS can have seasonal batch-to-batch variations which can further contribute to issues in reproducibility.	[30]
Advanced DMEM/F12	Rich in nutritional factors, such as glucose, amino acids, vitamins, zinc, putrescine, hypoxanthine, and thymidine. It is usually coupled with FBS due to its lack of proteins and growth factors.	[5]
HEPES	A non-volatile buffer is used to maintain a stable pH while culturing. It is especially useful as a buffering agent when cells are required to be outside of the incubator for extended periods.	[36]
GlutaMAX	An exact substitute for L-glutamine, which has been demonstrated to be an effective nutrient for cancer cells to provide more nitrogen and carbon for biosynthetic processes, supporting unchecked proliferation of cancer cells.	[30,54]
Hydrocortisone	Helps support cell viability and proliferation, maintain hormone sensitivity, modulate cellular responses, and enhance experimental consistency. Note: Dose-dependent cytotoxic effects have been reported, inhibiting proliferation and inducing cell cycle arrest.	[55]
Human Epidermal Growth Factor (hEGF)	hEGF is used to promote cell proliferation by binding to the EGF receptor on the cell surface, maintaining cell viability, and enhancing the responsiveness of 3D models to drugs/therapeutic agents. Note: Although elevated hEGF may enhance proliferation when coupled with BME (Matrigel/Culturex), it has been correlated to hindering the structural integrity of the organoid with gradual sinking and 3D organization loss.	[47]
Antibiotics	Antibiotics, such as Pen/Strep and Gentamicin, are used to help prevent cell contamination. They either act by inhibiting cell wall synthesis or interfering with membrane permeability. Note: Regular antibiotic usage may result in bacterial contamination resistance, which might impact cell proliferation and differentiation. It can also drastically change the regulation and expression of genes, which might change the outcomes of research on medication response, cell regulation, and differentiation.	[56]

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
