# Peer review of "The Transformative Role of 3D Culture Models in Triple-Negative Breast Cancer Research"

_cancers, 2024, doi:10.3390/cancers16101859_

Round 1

Reviewer 1 Report

Comments and Suggestions for Authors

This manuscript highlights the role of 3D Culture Models advances in cancer research, especially in TNBC, giving special attention to PDO and PDXOs. Additionally, it proposes to overview diverse 3D culture models derived from cell lines and tumors highlighting their advantages and culture challenges. It also discusses various culturing methodologies and imaging techniques essential for advancing this critical area of research.

This manuscript is extremely relevant to the field of breast cancer research, especially TNBC. It is generally quite well-written, with recent literature, and covers a wide broad of topics in this research area. This type of review is still scarce in the literature, so I believe this manuscript deserves publication. Below I leave some comments and suggestions that I believe could be improved to increase the impact and the relevance of the manuscript to researchers in this field.

Comments:

1) Material and Methods: You should include the number of studies included in this review after using the. keywords and selecting the ones that emphasize the 3D cancer models in TNBC research.

2) Figures, in general, could be more described, directing the reader to the different aspects of the Figure. As a reader, I had some difficulties in following the sequence of the text with what is described in the figures. For instance, Figure 2, could have a letter code a), b).. and in the text instead of saying just Figure 2, could be Figure 2a, for the hanging drop system, Figure 2b for magnetic levitation, and so on…. Figure 2 must have been referred to in all 3.2 topics.

3) For better understanding and clear information assessment, a table with the advantages and challenges of the different 3D cell culture methods in BC would be very useful. (This is an optional suggestion, but I could not leave this suggestion away from this revision). Of particular interest should be addressed the differences, advantages, and challenges of PDO and PDXO models.

4) References position on the text: In several parts of the text, the references seemed to be included not in the correct part of the paragraphs. For some examples: in lines 134-136, the limitations the authors refer to were described in the references mentioned in the sentence before that finishes also in line 134. If this is so, I believe these references should be at the end of the paragraph. Similar to lines 191-194; 229-231; 247-249; 256-258; 348-356; 356-359; 361-366; 399-406 and other similar situations.

5) This manuscript proposes an overview of 3D culture models derived from cell lines and tumors highlighting the advantages and challenges, I felt that in some parts of this manuscript, it was more generalist and failed to put the spot more on TNBC. Try to call more attention to the literature Also the manuscript dedicates one section to endpoint assays, I missed the histological and immunohistochemical techniques on this topic. Please include some studies where these techniques made an important contribution to this research area.

6) The transformative role of various 3D cancer models in elucidating TNBC pathogenesis and guiding novel therapeutical strategies proposed in the objectives was not directly related to the Discussion and Conclusions sections, where there is described a general overview of 3D models not specifying Breast cancer and more specifically TNBC as described in the title and the objectives of the manuscript.  Please, review both sections and try to be more specific relating to the objectives of this review.

7) The References list has several incongruencies related to the citation style, like dates, volumes, and pages. Some are not right at all like Ref [25], Ref [58]. All references should be revised and corrected to fulfill all requirements of the journal rules.

Author Response

Dear reviewer,

I am writing to express my sincere gratitude for the time and effort you dedicated to reviewing my review article, “The Transformative Role of 3D Culture Models in Triple-Negative Breast Cancer Research”, submitted to the journal Cancers. The insightful comments and constructive suggestions you provided have immensely enriched the quality and depth of the manuscript.

In response to your suggestions, I have carefully revised several sections of the manuscript to address the concerns you raised and to incorporate the additional perspective you provided. I would like to highlight some specific areas where your feedback has been instrumental in refining the content:

General adjustments

  • We rewrote the “Discussion” and “Conclusion” sections to emphasize the significance of 3D culture models in Triple-Negative Breast Cancer research, aligning with the article’s objectives (lines: 565 - 596, and 598 – 617, respectively).
  • To improve clarity, we provided detailed figure legends, including letter codes, and introduced a new table addressing the advantages and challenges of 2D and 3D cancer models (Table 1, line: 160).
  • References have been carefully adjusted to meet journal requirements for positioning and style.

Specific suggestions

  • “Material and Methods: You should include the number of studies included in this review after using the keywords and selecting the ones that emphasize the 3D cancer models in TNBC research.”

Approximately 34 studies emphasizing the application of 3D cancer models in Triple-Negative Breast Cancer research were selected after our PubMed searches.

  • “The manuscript dedicates one section to endpoint assays, I missed the histological and immunohistochemical techniques on this topic. Please include some studies where these techniques made an important contribution to this research area.”

We included a methodology for integrating histological and IHC techniques on 3D culture models, with references to relevant studies. We also incorporated two new studies demonstrating the use of artificial intelligence and computational methods for organoid morphological analysis.

Please find attached the revised version of the manuscript. We hope these modifications are appropriate and that this manuscript is now acceptable for publication.

X. Bittman

Reviewer 2 Report

Comments and Suggestions for Authors

A review manuscript titled, "The Transformative Role of 3D Culture Models in Triple-Negative Breast Cancer Research" by Xavier et al., described the TNBC biology comprehension and enhancing drug response predictions focussing on patient-derived organoids and multi-omics analyses. The manuscript is well organized and presents comprehensive information about the topic of interest. Following minor revisions are to be implemented before the acceptance of the manuscript:-

1. In acknowledgement, Figures are generated using some software. Hence, the authors need to mention the source of generating the figures in the legends of each. 

2.  A concise table can be provided to discuss drugs discovered using 3D cancer models.

Author Response

Dear reviewer,

I am writing to express my sincere gratitude for the time and effort you dedicated to reviewing my review article, “The Transformative Role of 3D Culture Models in Triple-Negative Breast Cancer Research”, submitted to the journal Cancers. The insightful comments and constructive suggestions you provided have immensely enriched the quality and depth of the manuscript.

In response to your suggestions, I have carefully revised several sections of the manuscript to address the concerns you raised and to incorporate the additional perspective you provided. I would like to highlight some specific areas where your feedback has been instrumental in refining the content:

General adjustments

  • We rewrote the “Discussion” and “Conclusion” sections to emphasize the significance of 3D culture models in Triple-Negative Breast Cancer research, aligning with the article’s objectives (lines: 565-596, and 598-617).
  • To improve clarity, we provided detailed figure legends, including letter codes, and introduced a new table (Table 1, line: 160) addressing the advantages and challenges of 2D and 3D cancer models.
  • References have been carefully adjusted to meet journal requirements for positioning and style.

Specific suggestions

  • “In acknowledgment, Figures are generated using some software. Hence, the authors need to mention the source of generating the figures in the legends of each.”

We ensured to include the source for generating each figure in their respective legends. Additionally, we highlighted that all of our figures are being published for the first time.

  • “A concise table can be provided to discuss drugs discovered using 3D cancer models.”

In this article, we aimed to comprehensively analyze the utilization of 3D cancer models, providing a detailed examination of their strengths, limitations, and their significant role in drug discovery and development. Despite our efforts, we encountered difficulty in generating a table specifically focused on drugs discovered using 3D cancer models without duplicating information already elucidated within the text.

Please find attached the revised version of the manuscript. We hope these modifications are appropriate and that this manuscript is now acceptable for publication.                          

Sincerely,

X. Bittman

Reviewer 3 Report

Comments and Suggestions for Authors

Overall, the review article provides a well-written and comprehensive overview of the transformative role of 3D culture models in triple-negative breast cancer (TNBC) research. It effectively captures the current state of research in the field. To further enhance the article:

  1. Comparative Analysis: It would be beneficial to include a comparative analysis between 3D cell culture models of TNBC and those of other organs or tissues. Highlighting similarities and differences in model complexity, biological relevance, and utility in drug screening or disease modeling would provide valuable insights.Exploring and discussing the research progress in organoid models of other cell types compared to TNBC organoids can add depth to the discussion.
  2. Alternative Approaches: Delving into alternative approaches or modifications to traditional 3D cell culture settings for TNBC organoids, if any available, could provide valuable insights into how culture conditions, growth factors, and matrix components impact model development and application.
  3. Positioning in Research Process: Clarifying the stage at which 3D culture models lie within the entire research process is crucial. Understanding what comes before (e.g., initial screening using 2D models) and after (e.g., incorporation of temporal dynamics in 4D models) 3D research, including their respective advantages and limitations, would contextualize the importance of 3D models in studying TNBC progression, metastasis, and therapeutic responses.

Author Response

Dear reviewer,

I am writing to express my sincere gratitude for the time and effort you dedicated to reviewing my review article, “The Transformative Role of 3D Culture Models in Triple-Negative Breast Cancer Research”, submitted to the journal Cancers. The insightful comments and constructive suggestions you provided have immensely enriched the quality and depth of the manuscript.

In response to your suggestions, I have carefully revised several sections of the manuscript to address the concerns you raised and to incorporate the additional perspective you provided. I would like to highlight some specific areas where your feedback has been instrumental in refining the content:

General adjustments

  • We rewrote the “Discussion” and “Conclusion” sections to emphasize the significance of 3D culture models in Triple-Negative Breast Cancer research, aligning with the article’s objectives (lines: 565 - 596, and 598 – 617).
  • To improve clarity, we provided detailed figure legends, including letter codes, and introduced a new table (Table 1, line: 160) addressing the advantages and challenges of 2D and 3D cancer models.
  • References have been carefully adjusted to meet journal requirements for positioning and style.

Specific suggestions

  • “Comparative Analysis: It would be beneficial to include a comparative analysis between 3D cell culture models of TNBC and those of other organs or tissues. Highlighting similarities and differences in model complexity, biological relevance, and utility in drug screening or disease modeling would provide valuable insights. Exploring and discussing the research progress in organoid models of other cell types compared to TNBC organoids can add depth to the discussion.”

We provided evidence from studies comparing TNBC organoid morphology with other PDO models (lines 105-109), as well as their aggressive behavior with other breast and colon cancer models (lines 364-368). Additionally, we referenced a study examining TNBC organoids at the single-cell level (lines 578-587).

  • “Alternative Approaches: Delving into alternative approaches or modifications to traditional 3D cell culture settings for TNBC organoids, if any available, could provide valuable insights into how culture conditions, growth factors, and matrix components impact model development and application.”

We added a new section titled “3D Culturing Challenges and Alternative Approaches” (Section 3.3.3, lines 350-370) to our manuscript. This section discusses two alternative methods for culturing TNBC organoids: Nayak et al.’s scaffold-based system and Jain et al.’s deterministic 3D culturing approach.

  • “Positioning in Research Process: Clarifying the stage at which 3D culture models lie within the entire research process is crucial. Understanding what comes before (e.g., initial screening using 2D models) and after (e.g., incorporation of temporal dynamics in 4D models) 3D research, including the irrespective advantages and limitations, would contextualize the importance of 3D models in studying TNBC progression, metastasis, and therapeutic responses.”

In the “Discussion” section (lines 568 to 576) we have provided clarification regarding the stage of the research process where 3D culture models are situated. This addition contextualizes their importance in studying cancer progression and therapeutic responses.

Please find attached the revised version of the manuscript. We hope these modifications are appropriate and that the manuscript is now acceptable for publication.                          

Sincerely,

X. Bittman
